# Hepatoprotective Mechanisms of Taxifolin on Carbon Tetrachloride-Induced Acute Liver Injury in Mice

**DOI:** 10.3390/nu11112655

**Published:** 2019-11-04

**Authors:** Chao-Lin Yang, Yu-Shih Lin, Keng-Fan Liu, Wen-Huang Peng, Chih-Ming Hsu

**Affiliations:** 1Ph.D. Program for Biotechnology Industry, College of Biopharmaceutical and Food Sciences, China Medical University, Taichung 40402, Taiwan; 2Department of Pharmacy, Chiayi Chang Gung Memorial Hospital, Chiayi 61363, Taiwan; yohimba@cgmh.org.tw; 3Department of Chinese Pharmaceutical Sciences and Chinese Medicine Resources, College of Chinese Medicine, China Medical University, Taichung 40402, Taiwan; cell77821@gmail.com; 4Department of Medical Education, Chiayi Chang Gung Memorial Hospital, Chiayi 61363, Taiwan

**Keywords:** liver injury, taxifolin, antioxidant enzymes, carbon tetrachloride, mice

## Abstract

Objective: To investigate the hepatoprotective mechanisms of taxifolin in mice with acute liver injury induced by CCl_4_. Methods: ICR (Institute of Cancer research) mice were orally pretreated using taxifolin for 7 consecutive days and were then given single intraperitoneal (i.p.) injections of 0.2% CCl_4_ (10 mL/kg body weight, i.p.). Liver injury was then determined using assays of serum alanine aminotransferase (sALT) and serum aspartate aminotransferase (sAST). Further, to investigate the hepatoprotective mechanisms of taxifolin, we determined malondialdehyde (MDA) levels and superoxide dismutase (SOD), glutathione peroxidase (GPx), and glutathione reductase (GRd) activities. Results: CCl_4_-induced liver injury led to significant increases in sALT and sAST activities, and these increases were limited by taxifolin and silymarin (Sily) pretreatments. Histological analyses also indicated that taxifolin and Sily decreased the range of liver lesions in CCl_4_-treated mice and vacuole formation, neutrophil infiltration, and necrosis were visibly reduced. In addition, SOD, GPx, and GRd activities were increased and MDA levels were decreased after taxifolin and Sily treatments. Conclusion: The hepatoprotective mechanisms of taxifolin and Sily are related to decreases in MDA levels presumably due to increased antioxidant enzyme activities. These outcomes suggest that taxifolin mitigates acute liver injury resulted from CCl_4_ in mice, demonstrating the hepatoprotective effects of taxifolin.

## 1. Introduction

The formation and degradation of reactive oxygen species (ROS) in all aerobic organisms is commonly known; further, ROS mediate various intracellular signaling cascades [1]. The excessive production of ROS induces oxidative stress in human bodies, causing damage to proteins, DNA, and lipids and leading to degenerative and pathological disease states [2]. Some environmental factors, such as cigarette smoke and certain drugs, can increase free radical activities in the liver. To a certain extent, endogenous antioxidants can maintain oxidative equilibrium and prevent cell damage from excess ROS under such conditions [3].

The liver plays an essential role in various metabolic processes; moreover, liver injury leads to various severe morbidities. Various toxic chemicals and drugs and infection with viruses, such as hepatitis, cause liver injury [4]. Carbon tetrachloride (CCl_4_) metabolism via cytochrome P450 leads to the formation of trichloromethyl radicals [4,5] that rapidly react with lipids and proteins, in particular, causing lipid peroxidation of cell membranes [6]. Previous research indicates that antioxidants inhibit the formation of free radicals and protect liver tissues against CCl_4_-induced injury via blocking deleterious lipid peroxidation reactions. To exploit these mechanisms, the antioxidant activities of various herbs and health foods have been comprehensively investigated in recent years, and numerous active chemical compounds have been identified [7].

Taxifolin is present at low levels in plants but has been identified as a bioflavonoid pseudovitamin P. Taxifolin is isolated from larch plants, such as larch and Douglas fir, in which it is present at 2–3%.

Taxifolin reportedly prolongs life by 40% and 37% in mice with leukemia [8]. Strong antibacterial activity has also been demonstrated against *Staphylococcus aureus*, *Escherichia coli*, *Shigella*, and *Salmonella typhi*. Moreover, the biological activities of taxifolin have been related to reduced free radical activities in humans and are associated with anti-aging and antitumor effects and inhibition of cytopathic effects [9]. Although the larch plant has been traditionally used to cure liver disease, the related mechanisms have not yet been scientifically validated. Therefore, we investigated the hepatoprotective antioxidant activities of taxifolin in CCl_4_-induced acute liver injury in mice. To this end, we monitored liver injury according to the levels of serum alanine aminotransferase (sALT) and serum aspartate aminotransferase (sAST) [10]. Histopathological changes were also observed in liver biopsies, and malondialdehyde (MDA) levels were determined along with antioxidant enzymes superoxide dismutase (SOD), glutathione peroxidase (GPx), and glutathione reductase (GRd) activities [11].

## 2. Materials and Methods

### 2.1. Animals

Male ICR mice (20–25 g) purchased from BioLasco Charles River Technology (Taipei, Taiwan) were used in the study. The animals were raised in a temperature-controlled room of the Animal Center of China Medical University at 22 ± 1 °C (relative humidity: 55% ± 5%; light and dark cycle: 12 h light/12 h dark) for at least one week. Mice were provided food and clean water ad libitum. All animal tests were performed as per the NIH Guidelines for the Care and Use of Laboratory Animals. Further, the Committee for Animal Research at China Medical University approved the experimental protocol (IACUC 2018–249).

### 2.2. CCl_4_-Induced Hepatotoxicity

Mice were randomly separated into 6 groups (*n* = 10). Mice in the control and CCl_4_ groups were orally administered distilled water, and those in the Sily group were orally administered Sily at 200 mg/kg in 1% carboxymethylcellulose [12]. Mice in the taxifolin group were orally administered taxifolin at 100, 150, and 200 mg/kg for 7 consecutive days. Further, 1 hour after the final pretreatment, CCl_4_ was intraperitoneally (i.p.) injected into all mice except those in the control group, which received equivalent volumes of olive oil (i.p.). Mice were euthanized under anesthesia at 24 h after CCl_4_ injections, and blood was collected, followed by centrifugation at 3000 rpm (Beckman GS-6R, Germany) at 4 °C for 30 min to separate the serum. Subsequently, sALT and sAST activities were measured by spectrophotometric diagnostic kits (Roche, Germany). Removal of liver tissues was performed for histological analyses, MDA assays, and measurements of antioxidant enzymatic activities.

### 2.3. Histological Analyses

Fresh liver tissues from the same lobes were collected, trimmed to a thickness of about 2 mm, and then fixed in 10% buffered formaldehyde solution. Fixed tissues were cut into sections of 2 µm and stained using hematoxylin and eosin (H&E) followed by examination under a light microscope (Olympus, Yuan Li Instrument Co., Ltd., Taichung Branch.).

### 2.4. MDA Assays

The levels of MDA were determined by thiobarbituric acid reacting substance method. Absorbance was determined at 532 nm, and the levels of MDA were expressed as μmol/mg protein [12].

### 2.5. Measurements of Antioxidant Enzyme Activity

SOD activities were determined as the procedure described by Misra and Fridovich [13]. Absorbance was measured at 480 nm for 4 min; one unit of SOD activity was expressed as the enzyme amount required for inhibiting epinephrine oxidation by 50%.

GPx activities were determined as the protocol by Flohe and Gunzler [14]. Absorbance was measured at 340 nm for 180 s, and a molar extinction coefficient of 6.22 × 10^−3^ was used to calculate enzyme activity. In these experiments, one unit of activity is defined as that needed to oxidize NADPH at a rate of 1 mM/min/mg protein.

As per the method by Carlberg and Mannervik [15], in which absorbance is measured at 340 nm for 3 min and a molar extinction coefficient of 6.22 × 10^−3^ was used to calculate product concentrations, GRd activity was determined. In these assays, one unit of activity is defined as that needed to oxidize NADPH at a rate of 1 mM/min/mg protein.

### 2.6. Statistical Analysis

Means ± standard errors of the mean (SEM) are used to present the data. SPSS software was used to perform statistical analyses. One-way ANOVA followed by Tukey–Kramer tests (Kramer’s Method) was used to identify significant differences. Quantitative analyses of histological data were performed using nonparametric Kruskal–Wallis tests followed by Mann–Whitney U-test. *p* < 0.05 was considered statistically significant.

## 3. Results

### 3.1. CCl_4_-Induced Acute Liver Injury

The hepatoprotective effects of taxifolin in mice with CCl_4_-induced liver injury are presented in Figure 1A,B. CCl_4_ treatments significantly increased the activities of sALT and sAST. However, pretreatments with taxifolin (200 mg/kg) and Sily (200 mg/kg) significantly reduced the elevated activities of sALT and sAST induced by CCl_4_ in mice.

### 3.2. Histological Analyses

As shown in Figure 2 and Table 1, CCl_4_ treatments led to the formation of vacuoles and necrosis in histological sections, and these indicators of tissue injury were reduced in animals that were pretreated with taxifolin (150 and 200 mg/kg) and Sily (200 mg/kg).

### 3.3. MDA Levels

In lipid peroxidation assays, MDA levels (Figure 3) were significantly increased in mice with liver injury induced by CCl_4_. Yet, in mice of the taxifolin and Sily groups, hepatic MDA levels mediated by CCl_4_ were significantly lowered.

### 3.4. Antioxidant Enzyme Activities

SOD, GPx, and GRd activities were significantly decreased in mice with CCl_4_-induced liver damage than in the control group (Figure 4). The decreases in these activities were significantly ameliorated via pretreatments with taxifolin (200 mg/kg) and Sily (200 mg/kg).

## 4. Discussion

Enhanced sAST and sALT activities are generally associated with hepatic structural damage [16]. Accordingly, compared with the control group, the present CCl_4_-treated mice had elevated sALT and sAST activities, and vacuole formation and extended necrotic areas around central veins indicated increased hepatic cell injury. Under these conditions, taxifolin pretreatments were significantly protective against histological and biochemical indicators of liver injury and CCl_4_-induced hepatotoxicity.

A major index of CCl_4_-induced liver injury is lipid peroxidation [6], and the endproduct MDA is widely used as a free radical-mediated lipid peroxidation marker [17]. Among contributors to the free radical milieu, NO is produced by l-arginine conversion to l-citrulline by inducible nitric oxide synthase (iNOS). Overproduced NO has been widely associated with inflammation and hepatic injury, and is abundantly implicated because it reacts with superoxide anions to form peroxynitrite, which acts directly as a lipid oxidant that elevates hepatic MDA levels in injured livers. In this study, hepatic MDA levels increased by CCl_4_ treatments were significantly decreased by pretreatment with taxifolin (0.5 and 1.0 g/kg). This result suggested taxifolin possessed potent hepatoprotective effects.

In a previous study, ROS after CCl_4_ treatments inactivated the antioxidant enzymes SOD, GPx, and GRd [17]. Superoxide anions are converted by SOD to the toxic intermediate H_2_O_2_ [18], further metabolized to nontoxic products by GPx. In this process, GSH is oxidized to GSSG and then recycled through GRd-mediated reduction to GSH [19]. Herein, we report significant decreases in SOD, GPx, and GRd activities following acute CCl_4_ damage. We also show that pretreatments with taxifolin significantly ameliorate these deleterious changes. Collectively, the present data suggest that taxifolin inhibits ROS production by promoting hepatic antioxidant activities, leading to greater protection against liver injury induced by CCl_4_.

## 5. Conclusions

In all the liver injury tests in this study, taxifolin exhibited hepatoprotective activities against acute liver toxicity induced by CCl_4_ in mice. Taxifolin’s hepatoprotective mechanisms should be associated with the inhibition of lipid peroxidation via the increase in the activities of antioxidant enzymes, such as GPx, SOD, and GRd, rather than a direct ROS scavenging effect of taxifolin. The present experiments provide evidence that taxifolin has hepatoprotective effects.

## Figures and Tables

**Figure 1 nutrients-11-02655-f001:**
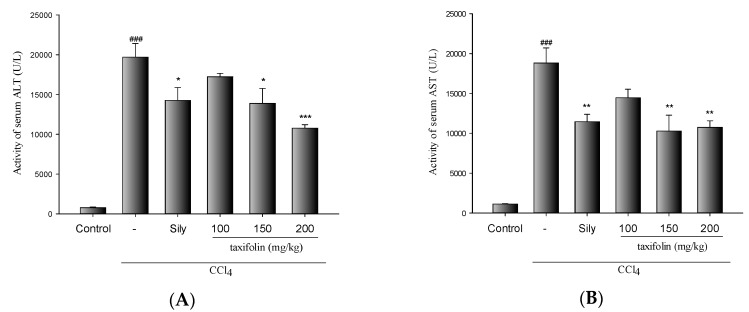
Effects of taxifolin and Sily on the activities of serum alanine aminotransferase (sALT) (**A**), and serum aspartate aminotransferase (sAST) (**B**) in carbon tetrachloride (CCl_4_)-treated mice; data are presented as means ± SEM (*n* = 10). ### *p* < 0.001, compared with the control group; * *p* < 0.05, compared with the CCl_4_ group. One-way ANOVA followed by Tukey–Kramer tests was used to identify the differences.

**Figure 2 nutrients-11-02655-f002:**
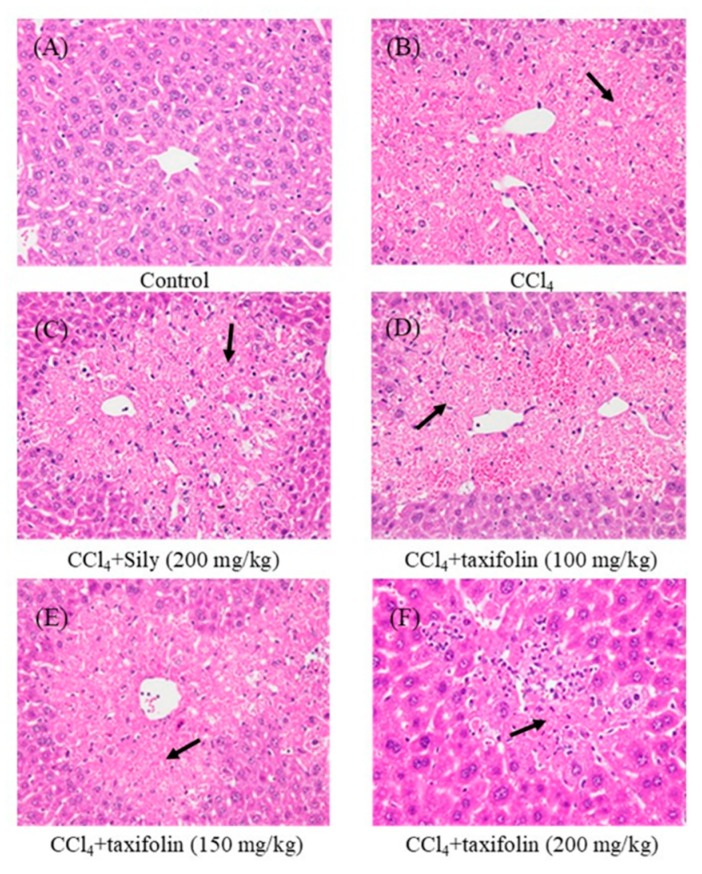
Histopathological changes in mice with acute hepatic injury induced by CCl_4_; images are presented from control and CCl_4_ treated mice with and without Sily and taxifolin pretreatments. Normal glycogen infiltration in a mouse of the control group (**A**); in liver sections from mice treated with CCl_4_, hematoxylin and eosin (H&E) staining showed hepatocellular vacuolization and moderate hepatic necrosis in portal areas (**B**). Sily (**C**) 100 mg/kg taxifolin (**D**), 150 mg/kg taxifolin (**E**), and 200 mg/kg taxifolin (**F**) pretreated CCl_4_-induced hepatic injury mice; 400×.

**Figure 3 nutrients-11-02655-f003:**
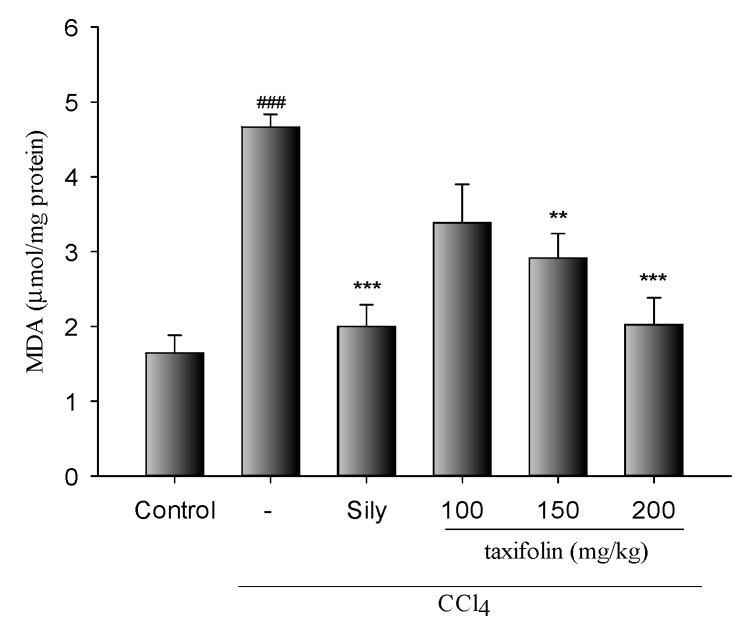
Taxifolin and Sily effects on hepatic MDA levels in mice with hepatic injury induced by CCl_4_; data are presented as means ± SEM (*n* = 10). ### *p* < 0.001, compared with the control group; *** *p* < 0.001, compared with the CCl_4_ only group; ** *p* < 0.05, compared with the CCl_4_-only group; One-way ANOVA followed by Tukey–Kramer tests was used to identify the differences.

**Figure 4 nutrients-11-02655-f004:**
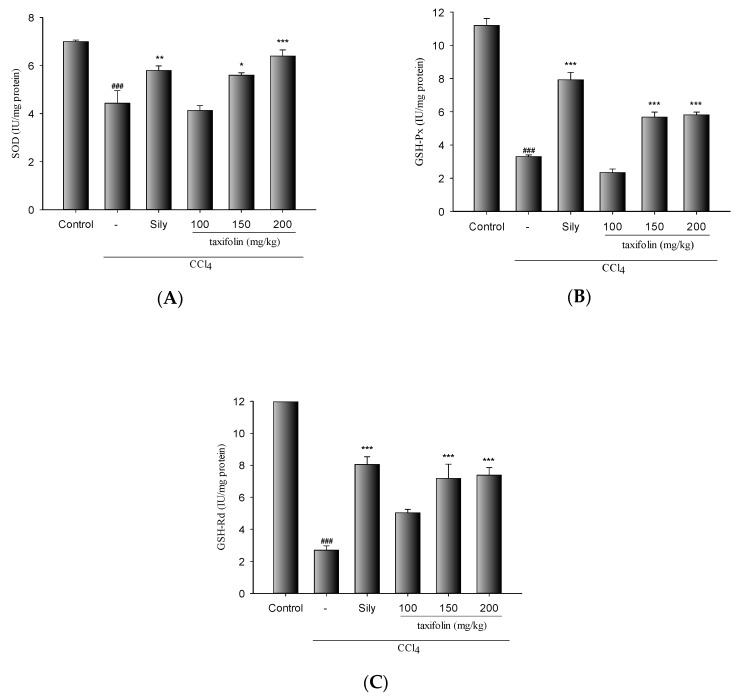
Taxifolin and Sily effects on the activities of hepatic superoxide dismutase (SOD) (**A**), glutathione peroxidase (GPx) (**B**), and glutathione reductase (GRd) (**C**) in CCl_4_-treated mice; data are presented as means ± SEM (*n* = 10). ### *p* < 0.001, compared with the control group; * *p* < 0.05 and ** *p* < 0.01, compared with the CCl_4_ group. One-way ANOVA followed by Tukey–Kramer tests was used to identify the differences.

**Table 1 nutrients-11-02655-t001:** Quantitative summary of the protective effects of Sily and taxifolin on hepatic damage induced by CCl_4_ based on histopathological observations.

Organ	Lesions	Group					
		N	CCl_4_	Sily	Taxifolin (mg/kg)		
					100	150	200
	Vacuolization^1^	0 ***	1.4 ± 0.18	0.8 ± 0.13 **	0.8 ± 0.14	1 ± 0.15 *	0.5 ± 0.17 ***
	Coagulative necrosis	0 ***	4	3.0 ± 0.31 *	3.9 ± 0.10	3.5 ± 0.17	2.6 ± 0.43 *

Lesions were graded from 1 to 5 as per the severity: 1 = minimal (<1%); 2 = slight (1–25%); 3 = moderate (26–50%); 4 = moderate/severe (51–75%); and 5 = severe/high (76–100%);.* *p* < 0.05, ** *p* < 0.01, and *** *p* < 0.001, compared with the CCl_4_ group.

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
