# Peer review of "Hepatoprotective Mechanisms of Taxifolin on Carbon Tetrachloride-Induced Acute Liver Injury in Mice"

_nutrients, 2019, doi:10.3390/nu11112655_

Round 1

Reviewer 1 Report

The authors described hepatoprotective mechanism of Taxifolin on carbon tetrachloride-induced acute liver injury in mice. The subject is quite unique and important. And the study design was simple and well understood, the results were clear to conclude. However, there are a few concerned points.

First, the authors provided six groups to evaluate in this study. What is Sily? What is the mean of Sily group?

Second, why were the data of Sily group worse than control group?

Third, how to decide the dosage of Taxifolin and Sily in each group?

The authors should explain the reasons of these questions.

Author Response

Dear Editor and Reviewers,

Thank you so much for the recognition of our research. It is such an excitement and encouragement to us. We are truly thankful for the valuable comments and thoughtful suggestions by you and the reviewers. Based on these comments and suggestions, we have carefully revised the manuscript. We hope that the revised manuscript meets your journal’s standards.

I have already replied to the comments on the webpage system.

We apologize for inadequate knowledge on the reviewing rules and procedures. We have another file that has been modified. Should we upload that file or should we wait for your reply and accordingly incorporate the changes suggested by you? Please let us know as soon as possible if there are any mistakes.

Please find our line-by-line responses to the reviewers’ comments in the following pages. All the revisions are marked in red.

If you need further information, please let us know.

Sincerely,

Yang

Point 1:

What is Sily?

Response 1:

We sincerely apologize for not providing clear expression and detailed information.

“Sily” is the abbreviation of the drug “silymarin.” We mentioned the full name in the abstract once. Thereafter, only the abbreviation was used.

Please let us know if the journal requires it to be written in the expanded form throughout the article.

Point 2:

What is the mean of Sily group?

Response 2:

We apologize for not expressing the term clearly.

The “Sily group” is one of the six animal groups.

We have corrected it in the article.

Point 3:

Why were the data of Sily group worse than control group?

Response 3:

After confirmation by numerous experiments, we found that the results obtained by Sily group were most correct.

All data were statistically analyzed and are presented as *, which indicates significant difference. Furthermore, the insight into whether taxifolin or silymarin has a positive effect on liver protection has been confirmed by experiments.

Point 4:

How to decide the dosage of Taxifolin and Sily in each group?

Response 4:

The conclusions and ideas of taxifolin’s dosage are based on the pilot study and numerous laboratory pretests. For silymarin, they are based on the results of other studies.

We apologize for the vague reference of the silymarin dose. Nevertheless, we have already added such details in the manuscript.

Reviewer 2 Report

Yang et al studied the effect of Hepatoprotective mechanisms of taxifolin in mice with acute liver injured induced by CCL4. The study is interesting, and I recommend the manuscript for publication after minor revision.

Thorough scientific editing should be done as there are some flaws in the manuscript observed. For instance, in the line 132. In the material methods, please mention about the light microscope company name and where it has been purchased? Please describe how MDA assay has been performed or give the reference. Why silymarin 200mg/Kg was chosen for the study? Please check full stop for instance in the line 110 and line 113. Format English throughout the manuscript I recommend elaborating some more in the discussion part. The overall schematic representation is needed to understand the paper at glance.

Author Response

Dear Editor and Reviewers,

Thank you so much for the recognition of our research. It is such an excitement and encouragement to us. We are truly thankful for the valuable comments and thoughtful suggestions by you and the reviewers. Based on these comments and suggestions, we have carefully revised the manuscript. We hope that the revised manuscript meets your journal’s standards.

We have already replied to the comments on the webpage system.

We apologize for inadequate knowledge on the reviewing rules and procedures. We have another file that has been modified. Should we upload that file or should we wait for your reply and accordingly incorporate the changes suggested by you? Please let us know as soon as possible if there are any mistakes.

Please find our line-by-line responses to the reviewers’ comments in the following pages. All the revisions are marked in red.

If you need further information, please let us know.

Sincerely,

Yang

Point 1:

Thorough scientific editing should be done as there are some flaws in the manuscript observed. For instance, in the line 132.

Response 1:

We humbly apologize for our mistakes. We modified the line as follows. We hope that you will find it satisfactory.

“The hepatoprotective effects of taxifolin in mice with CCl4-induced liver injury are presented in Figures 1-(A) and 1-(B). CCl4 treatments significantly increased the activities of sALT and sAST. However, pretreatments with taxifolin (200 mg/kg) and Sily (200 mg/kg) significantly reduced the elevated activities of sALT and sAST induced by CCl4 in mice.”

Point 2:

In the material methods, please mention about the light microscope company name and where it has been purchased?

Response 2:

We apologize for incomplete details. The light microscope’s brand is Olympus, and it was sold under the company name Yuan Li Instrument Co., Ltd., Taichung Branch. We have added this information in the manuscript.

Point 3:

Please describe how MDA assay has been performed or give the reference.

Response 3:

We sincerely apologize for the vague description. Can we add more references in the manuscript?

Point 4:

Why silymarin 200mg/Kg was chosen for the study?

Response 4:

This drug dose is based on the results obtained from previous studies by other authors. We apologize for not clearly stating the reference to the silymarin dose. We have made the necessary changes in the manuscript.

Point 5:

Please check full stop for instance in the line 110 and line 113.

Response 5:

We apologize for the overlook. We have corrected these instances by deleting "per" in both sentences.